# The Complexity of the Post-Burn Immune Response: An Overview of the Associated Local and Systemic Complications

**DOI:** 10.3390/cells12030345

**Published:** 2023-01-17

**Authors:** H. Ibrahim Korkmaz, Gwendolien Flokstra, Maaike Waasdorp, Anouk Pijpe, Stephan G. Papendorp, Evelien de Jong, Thomas Rustemeyer, Susan Gibbs, Paul P. M. van Zuijlen

**Affiliations:** 1Department of Plastic Reconstructive and Hand Surgery, Amsterdam Movement Sciences (AMS) Institute, Amsterdam UMC, Location VUmc, 1081 HZ Amsterdam, The Netherlands; 2Department of Molecular Cell Biology and Immunology, Amsterdam Infection and Immunity (AII) Institute, Amsterdam UMC, Location VUmc, 1081 HZ Amsterdam, The Netherlands; 3Burn Center and Department of Plastic and Reconstructive Surgery, Red Cross Hospital, 1942 LE Beverwijk, The Netherlands; 4Association of Dutch Burn Centres (ADBC), 1941 AJ Beverwijk, The Netherlands; 5Department of Oral Cell Biology, Academic Centre for Dentistry Amsterdam (ACTA), University of Amsterdam and Vrije Universiteit Amsterdam, 1081 HZ Amsterdam, The Netherlands; 6Intensive Care Unit, Red Cross Hospital, 1942 LE Beverwijk, The Netherlands; 7Department of Dermatology, Amsterdam UMC, Location AMC, 1105 AZ Amsterdam, The Netherlands; 8Paediatric Surgical Centre, Emma Children’s Hospital, Amsterdam UMC, Location AMC, 1105 AZ Amsterdam, The Netherlands

**Keywords:** burn, immune response, inflammation, complexity

## Abstract

Burn injury induces a complex inflammatory response, both locally and systemically, and is not yet completely unravelled and understood. In order to enable the development of accurate treatment options, it is of paramount importance to fully understand post-burn immunology. Research in the last decades describes insights into the prolonged and excessive inflammatory response that could exist after both severe and milder burn trauma and that this response differs from that of none-burn acute trauma. Persistent activity of complement, acute phase proteins and pro- and anti-inflammatory mediators, changes in lymphocyte activity, activation of the stress response and infiltration of immune cells have all been related to post-burn local and systemic pathology. This “narrative” review explores the current state of knowledge, focusing on both the local and systemic immunology post-burn, and further questions how it is linked to the clinical outcome. Moreover, it illustrates the complexity of post-burn immunology and the existing gaps in knowledge on underlying mechanisms of burn pathology.

## 1. Introduction

Burn wounds are a significant problem worldwide, being the fourth most common type of trauma and ranking in the top 15 leading causes of the burden of disease globally [1]. Depending on the severity of the injury, burns can lead to sepsis, single or multiple organ failure and even death, and in the long-term, it can lead to problematic scarring with physical, psychological and social consequences [2,3]. Severe burns often induce a massive and long-term immune response both locally in the burn wound and systemically, that not only can negatively affect the wound healing but may also result in a systemic long-term impact on multiple organ systems [4]. 

In general, burn wound healing includes overlapping phases: the inflammatory phase, proliferation phase, and remodelling phase (Figure 1A–C). Interference with the natural course of these phases can result in adverse clinical outcomes [5]. 

For instance, several studies demonstrated dysregulated inflammation post-burn coincides with hypertrophic scar (HTS) formation [11,12] (Figure 1D), as well as systemic complications, e.g., hypermetabolism and organ dysfunction [13]. Although major improvements in acute burn care have decreased mortality rates, long-term consequences of burn trauma, such as locally delayed wound healing, pathologic scarring, secondary deepening, systemic thrombosis, sepsis, and endocrine and metabolic effects, are still difficult to treat. In order to predict the clinical outcome, develop more effective therapies, and reduce the burden of burn-related consequences, it is of utmost importance to unravel the pathophysiology of burns. In the last decades, it has been shown that severe burn trauma causes massive inflammation, which influences the local and systemic physiology, and persists for months up to years after injury [14,15,16,17]. Despite the detailed data about, e.g., cellular and molecular processes, little progress has been made in understanding and treating this exaggerated and prolonged inflammation after burns.

The main objective of this “narrative” review is to give an overview of the current state of knowledge of the immune response after burn injury and related clinical complications.

## 2. The Post-Burn Immune Response

### 2.1. The Innate Immune Response after Burn

#### 2.1.1. Acute Phase Response (APR) after Burn

Wound healing in the skin starts with the haemostasis and inflammation phase to restore haemostasis, eliminate invading pathogens, and remove necrotic tissue (Figure 1A). The initial cascade of events in the inflammatory phase starts when damaged skin cells become necrotic and release a range of damage-associated molecular patterns (DAMPs), which become free after injury. Together with signals from pathogens that invade through the injured skin barrier, known as pathogen-associated molecular patterns (PAMPs), they contribute to the activation of the acute phase response (APR) of wound healing, which is characterised by a rapid increase in inflammatory mediators and activation of innate immune cells [6].

Activation of mast cells by immunoglobulin E, toxins or activated complement, results in rapid release of their granules containing cytokines, growth factors, histamine, bradykinin, cathepsins and proteases. Mast cell degranulation in the burn wound leads to locally increased vascular permeability, which is mainly mediated by histamines and bradykinin, enabling the invasion of systemic immune cells to the wound. However, increased vascular permeability also contributes to the dehydration of the patient [18].

In response to these signals, skin-resident cells, e.g., mast cells and Langerhans cells, become activated and release inflammatory cytokines, which in turn activate other immune cells that are needed for the elimination of pathogens and removal of necrotic tissue. Furthermore, keratinocytes, fibroblasts and endothelial cells at the site of injury release cytokines and growth factors that attract and activate immune cells as well [7,8,9]. Interleukin 6 (IL-6) and interleukin 8 (IL-8), also known as CXCL8, are significantly increased after days 1–4. Major systemic functions of IL-6 are stimulation of acute phase protein (APP) synthesis in the liver, induction of naïve T cell differentiation and promotion of angiogenesis [19]. IL-8 is important for neutrophil recruitment to the site of burn injury and has a function in the tissue remodelling phase as well. IL-6, in particular, is a very important mediator of the overall acute change in systemic concentration of cytokines, growth factors and APPs, i.e., the APR, which develops within hours post-burn [10]. The liver is the main organ that produces APPs, such as C-reactive protein (CRP), serum amyloid A, haptoglobin, fibrinogen, protein S and complement [20,21]. The fastest- responding APP is CRP, which is an important initiator of inflammation and one of the inducers of the complement cascade.

Neutrophils are the first line of immune cells that migrate into the wound (Figure 2). For this, IL-8 is of utmost importance. IL-8 is released by several cell types, e.g., epithelial cells, endothelial cells and macrophages. Neutrophils contribute to pathogen elimination via key mechanisms such as phagocytosis of opsonised pathogens, the release of reactive oxygen species (ROS), and neutrophil extracellular trap (NETs) formation [22,23,24]. Furthermore, neutrophils release inflammatory cytokines, e.g., IL-1 and tumour necrosis factor α (TNF-α), which in turn recruit monocytes from the blood [25].

#### 2.1.2. A Prolonged and Over-Active Systemic APR after Burn Injury: A Pivotal Role for Complement

Upon burn trauma, the release of large amounts of inflammatory mediators from the wound site can activate systemic immune cells that subsequently produce inflammatory mediators as well. Various studies on the post-burn immune response show alterations in blood concentrations of various cytokines, growth factors, and proteins post-burn (Figure 3), correlating with the extent of injury [27,28]. 

As such, pro-inflammatory cytokines IL-6 and IL-8 are significantly increased after day 1–4 up to months or even years post-burn in the blood of burn patients, correlating with the % TBSA burned [16,29,30,31,32,33,34,35,36]. CRP levels in the blood increase immediately after the burn and remain elevated for months [37]. Next, factors of the complement system are a main part of the APR post-burn [17]. In several studies, complement factor C3 in blood started to increase a few days post-burn and remained elevated for the entire study period, which could be for weeks [38] or months [23,33]. Another important complement factor, C4, was also elevated in the blood of a pig burn wound model, but for a shorter period and with a later concentration peak than C3. Importantly, after normalisation of both complements C3 and C4 levels locally in the burn wound, persistently elevated complement blood levels were found. This suggests that an extended systemic inflammatory response exists post-burn, in which complement plays an important role [23]. For some APPs, an acute decrease in blood concentration after burn trauma has been observed prior to the long-term increase [17]. For instance, complement factor C3 in blood initially decreases after burn injury, prior to an increase in C3 blood concentration [33]. Explanations for acute decreases in APP concentrations, in general, are increased permeability of local blood vessels, increased APP turn-over rate, and a decrease in APP production [16]. 

Anti-inflammatory IL-10 peaks at day one post-burn and declines thereafter, whereby peak concentrations correlate with the % TBSA burned and depended on the presence of sepsis. However, the net balance is in favour of the pro-inflammatory cytokines IL-6 and IL-8 [29,30,32,33,34,35,36]. 

TNF-α, a growth factor of major importance in the early systemic response post-burn, is mainly produced by macrophages, neutrophils and mast cells. This pro-inflammatory mediator has a wide range of partly conflicting effector immune functions. It has been associated with the cytotoxicity of damaged cells, with changes in lipid metabolism, but also with stimulation of immune cells and cell-mediated immunity [39]. Since cell-mediated immunity is important for the elimination of pathogens, the latter function could explain why higher TNF-α levels have been related to sepsis post-burn [40,41,42]. 

IL-1β is another pro-inflammatory mediator involved in acute inflammation at the burn wound site that is systemically elevated as well [16,31]. Again, variations in concentrations might be due to differences in detection limits, but there are also indications that enhanced IL-1β levels mainly exist in the tissue of the lungs and central nervous system [30,36]. 

IFN-γ is an important factor in the innate immune response, which has been shown to be elevated in the blood of burn patients [16,31,43]. However, cells of the adaptive immune system are a major source of IFN-γ as well, specifically the Th1 cells. The main cell types of the innate immune system that produce IFN-γ are NK cells, macrophages, and antigen-presenting cells. This cytokine has various functions, among others, the induction of macrophage activity [44].

### 2.2. The Adaptive Immune Response after Burn

Next to the innate immune system, the adaptive immune system is part of the systemic inflammatory response post-burn. Several T cell lineages and subsets, as well as B cells, play a role in local burn wound inflammation and the regulation of the proliferation phase. The acute phase is dominated by a local Th1 cytokine environment and shifts towards a mixed Th2/Th17 environment in the days following burn injury [45]. The systemic post-burn activity of T and B cells shows similarities with the local response to burn. The activity of T cells is integrated with other post-burn immunological alterations, such as the enhanced activity of certain innate immune and endocrine factors. Nitric oxide, for instance, can impair splenic T cell proliferation and reduce the production of the Th1 cytokines IFN-γ and IL-2, resulting in a shift towards a Th2 cytokine environment, which has been found in burn-injured mice [45,46,47,48]. In a similar way, stress hormones (e.g., norepinephrine and cortisol), which are often present in increased concentrations in burn victims, promote a Th2 cytokine environment by inhibition of Th1 cells and stimulation of Th2 cells [49]. Further, activated mast cells release Th2-stimulating cytokines as well [50]. Moreover, this dominance of Th2 cells is consistent with the previously reviewed studies on plasma cytokine levels, of which the majority found significant increases in plasma levels of IL-10 in burn patients. Thus, the dominance of the Th2 phenotype is associated with an anti-inflammatory cytokine milieu, with a relative abundance of, amongst others, IL-10. This could contribute to an immunosuppressive condition post-burn since IL-10 is known for its stimulation of regulatory T cells, which downregulate the activity of effector T cells [46]. 

Another subset of αβ T cells that might contribute to altered systemic immunity is Th17 cells. Elevated levels of the Th17 cytokine IL-17 were found in burn wounds in the early phase post-injury. Similarly, some studies have observed systemic increases in IL-17 for a sustained period of time in patient blood [16]. Since Th17 cells are involved in the immunity of mucosal and epithelial linings, changes in their activity post-burn might result in systemic infections [46]. However, further research is needed to investigate the systemic effects of Th17 cells and the time course of Th17 activity after burn injury.

Next to the αβ T cell types, γδ T cells types seem to play a role in the post-burn systemic alterations as well. The γδ T cells are important in burn wound healing since they influence the local balance of the pro- and anti-inflammatory cytokines at the wound site. Their activity could alter systemic cytokine levels as well, thereby influencing the behaviour of other immune cells, e.g., recruitment of neutrophils to organs beyond the injured skin, such as the lungs, where they could induce tissue damage [51]. 

Relatively little research has been reported on the role of B lymphocytes in the systemic immune response to burn. Several studies have followed antibody levels over time in the blood of burn patients and found decreased levels of IgM, IgA and IgG compared to healthy controls, which correlated with the degree of burn depth, but not with the burn wound size [52,53,54]. The acute decline in antibody levels might be attributed to a combination of extravasation of antibodies to local burn wound fluid, lowered antibody production, and higher catabolism [54]. 

The activity of factors that are normally involved in the adaptive immune response is remarkable since they are normally linked to delayed response or skin memory, while in burns, most of the studies have shown that the acute inflammation response is disordered. However, the activity of these factors can still be linked to innate immunity function, e.g., activated Th17 cells can directly recruit human neutrophils, which is one of the acute inflammatory cells, via endogenous IL-8 [55]. Unfortunately, so far, there are no studies on burns that can provide a definitive answer.

## 3. Local Complications Related to a Dysregulated Immune Response after Burn Injury

### 3.1. Burn Wound Deepening

One of the local outcomes of burns can be wound expansion depth. This is generally known as burn deepening, the transition from a superficial partial-thickness wound to a deep partial- or full-thickness wound [56]. The fundamental concept of the burn wound anatomy originates from Jackson [57], who described the three well-known burn wound zones, namely the zone of coagulation, the zone of stasis, and the outer zone of hyperaemia. The zone of coagulation is the primary site of injury, the site of the most damage and will rapidly undergo necrosis. Outside the zone of coagulation is the zone of stasis, which is characterised by reduced blood flow (i.e., ischemia). The established theory is that the zone of stasis is crucial in the pathophysiology of burn wound deepening and that this zone can either increase viability or further decrease perfusion and become necrotic within 72 h [58]. The zone of hyperaemia is the outer zone, where microvasculature is not damaged but displays increased blood flow and inflammation. 

#### Immunological Pathways in Burn Wound Deepening

Key immunological factors that contribute to burn wound deepening are an enhanced inflammatory response with increased oxidative stress, microvascular thrombosis, apoptosis and autophagy [56,58,59]. The inflammatory response is essential for wound healing; however, persistent inflammation could result in increased local damage, eventually leading to burn wound deepening. A relation exists between local complement persistence and the presence of inflammatory cell infiltrates [14,23]. This suggests that the complement cascade contributes to excessive inflammation, thereby mediating burn wound deepening. Neutrophils, as one of the local infiltrating immune cells, could cause local damage to the microvasculature and surrounding tissue via the release of pro-inflammatory cytokines [60] and probably via NETs [59]. Although no direct causal relationship between the release of NETs and the initiation of microvascular thrombosis post-burn has been shown until now, an important (contributing) role for NETs in thrombosis is shown since NETs exert highly pro-coagulant effects after burn [59]. Thrombosis of the microvasculature in the burn wound highly contributes to hypo-perfusion in the burn wound and is a well-known mechanism in secondary wound deepening [58]. In addition to thrombosis, destructed endothelium in the zone of stasis may lead to fluid shifts and, thereby, hypo-perfusion of the zone of stasis. Factors that cause damage to the microvasculature are, amongst others, ROS, NOS and high thermal energy. Also, burn-related undernutrition may lead to local hypo-perfusion since a balanced electrolyte status is important for appropriate fluid distribution [56]. 

Lastly, both apoptosis and autophagy may contribute to burn wound deepening. As such, a higher rate of apoptosis of dermal cells in deep partial-thickness burns has been found compared to the non-burned skin of the same patient [61]. However, a more recent role in burn wound progression has been ascribed to autophagy, which is a mechanism by which, often stressed, cells deliver non-functional or unneeded cellular compartments to a phagosome for degradation or recycling. In this way, autophagy can be seen as a survival mechanism and could offer protection against apoptosis [20].

### 3.2. Hypertrophic Scar Formation

In the majority of burn patients, the remodelling phase results in a pathological scar [3]. The remodelling phase involves the contraction of the granulation tissue and reorganisation of the ECM; for this, it requires a balance in ECM production, wound contraction and activity of MMPs and tissue inhibitors of metalloproteinases (TIMPs) (Figure 1C). A major impairment for burn patients is the formation of hypertrophic scars, which could also lead to contractures. An illustration of a hypertrophic scar that summarises the risk factors and potentially involved biological mechanisms is depicted in Figure 1D. Although hypertrophic scars can develop after non-burn trauma as well, the prevalence is higher among burn patients [62]. However, the exact reason and mechanism behind this are not known yet. Moreover, there is a large variation in data on incidence rates of hypertrophic scar formation. Reported incidence rates post-surgery range from 39% to 68%, whereas hypertrophic scars incidence rates post-burn range from 33% to 91% [63]. Hypertrophic scars are raised and often itchy and painful; other main characteristics are redness and rigidness. Biological features of hypertrophic scars are the excessive presence of ECM, a decreased collagen I/III ratio with a parallel orientation of the collagen fibrils, and increased cellularity with the presence of myofibroblasts and fibrocytes. Also, enhanced reepithelisation is an aspect of hypertrophic scars [3,64,65]. Since burn trauma could affect large percentages of the body, scars, specifically hypertrophic scars, can be both functionally and aesthetically invalidating.

#### Immunological Pathways in hypertrophic Formation

The inflammatory environment has a regulatory role in the process of remodelling and hypertrophic scar formation [3,66,67] via paracrine interactions between dermal (myo)fibroblasts and immune cells, including mast cells, macrophages and T cells [68]. For example, high concentrations of TGF-β could prevent myofibroblasts apoptosis and induce myofibroblasts formation/differentiation, resulting in uncontrolled wound contraction and excessive production of ECM [69]. Also, the degree of (myo)fibroblast activity in burn patients depends on which Th cell phenotype dominates the environment. Th1 cells mainly release IFN-γ and IL-12, thereby downregulating (myo)fibroblast activity. In contrast, Th2 cells primarily produce cytokines that stimulate (myo)fibroblast activity, namely IL-4, IL-5, IL-10 and IL-13 [3,70]. Another paracrine actor is the mast cell, which could also modulate the activity of (myo)fibroblasts [26]. Although the exact mediator(s) is unknown, several studies have shown that mast cell proteases have fibrinogenic effects. For e.g., tryptase stimulates procollagen mRNA synthesis, contraction and differentiation of dermal fibroblasts into myofibroblasts. Histamine, another mast cell mediator, can also increase collagen production in fibroblasts [71,72]. 

Skin injury induces the expression of MMPs by inflammatory cells, keratinocytes and (myo)fibroblasts, which produce different types of MMPs. The main cytokine that promotes MMP transcriptional pathways is IL-1 [65]. In addition, MMPs help reorganise the ECM via the degradation of ECM components, such as collagen, fibronectin and elastin [73]. For instance, the degradation of collagen results in a change in the ratio between collagen types I and III [74]. Table 1 shows an overview of the local events after the burn based on the major “actors” and their actions. 

Several studies have shown that the early phase of wound healing could already contribute to hypertrophic scar formation [75,76]. In the first wound healing phase, namely haemostasis, disturbances in platelet activity may contribute to the development of hypertrophic scars. Since platelets release the fibroblast-stimulating platelet-derived growth factor (PDGF), overactivity of platelets could lead to hypertrophic scar formation [76]. Next, an abnormal course of the early inflammation phase in non-burn wounds could increase the risk of hypertrophic scar formation. However, opinions and findings on this are divided. A more common theory about the relationship between inflammation and hypertrophic scars is that the often prolonged or abundant inflammatory response to burn injury eventually results in hypertrophic scarring [10,21,76]. Though, another finding implicates that cytokine levels of the pro-inflammatory cytokines IL-6, IL-8, and CCL2 were decreased in wounds of patients with hypertrophic scar three hours after surgery, compared to the non-hypertrophic scar group. This correlation between low cytokine levels and hypertrophic scars was only found in the local environment; the cytokines levels in blood were not lower in the hypertrophic scar group than in the non-hypertrophic scar group [75]. In each case, it seems clear that the immune response is disrupted after the burn, which could contribute to hypertrophic scar formation. 

As such, in the last decades, research on post-burn hypertrophic scar formation has focused on a range of possibly involved aspects of the immune system. These include chemokine signalling [77], M1 and M2 macrophages [10,78,79], Th1 and Th2 phenotypes [70,80], the responsiveness of (myo)fibroblasts and fibrocytes [81,82,83], keratinocyte-fibroblast and mast cell-fibroblast interactions [84,85,86] and adipose-derived stem cells (ADSCs) [87]. 

Important in hypertrophic scar formation is the Th1/Th2 ratio in the remodelling phase post-burn, as relatively large numbers of Th1 cells result in an anti-fibrotic cytokine environment, whereas the dominance of Th2 cells enhances scarring [80]. A recent theory states that of main influence in fibrosis of the skin is the Th2 cytokines IL-4 and IL-13, which stimulate the same transcription pathways that eventually result in enhanced ECM production and high concentrations of TGF-β [70]. Another effect of Th2 dominance that has been described is the stimulation of T helper 3 (Th3) cells, which are also a source of the myofibroblast stimulating cytokine TGF-β [3]. The role for TGF-β producing T cells in hypertrophic scars post-burn has been supported since higher numbers of TGF-β producing T helper cells were found in hypertrophic scars than in normal, mature scars [88]. Next, the tissue-repair stimulating macrophage subset M2 also is a major source of TGF-β. Although the role of macrophages in hypertrophic scars post-burn is not fully understood yet, enhanced presence of macrophages or increased expression of TGF-β by macrophages could promote (myo)fibroblast activity and thus cause hypertrophic scar formation [10,76]. 

In addition to the TGF-β-rich environment after burn, fibroblasts in hypertrophic scars may respond differently to TGF-β than fibroblasts in mature scars. Deep dermal fibroblasts produce more ECM, TGF-B and CTGF and fewer MMPs than the more superficial fibroblasts [82,89], thereby promoting excessive scarring [90]. This likely is part of the explanation for the higher incidence of hypertrophic scars after deep partial-thickness and full-thickness burns than after superficial burns. Next, fibroblasts in HTS also showed more resistance to apoptotic signals than fibroblasts in non-hypertrophic scars [81]. 

## 4. Systemic Complications Related to a Dysregulated Immune Response after Burn Injury

### 4.1. Systemic Complications

The effect of burn injury on the human body does not end at the margins of the wound and/or by the time the wounds are closed. In fact, acute burn injury could have a severe systemic impact for years or even longer. Severe burn injuries induce a systemic response that is described as the systemic inflammatory response syndrome (SIRS) and is caused by an “over-exuberant” acute phase inflammation [91,92,93,94]. The excessive systemic release of pro-inflammatory cytokines, chemokines, lipids and vasoactive mediators leads to distant organ damage and multiple organ dysfunction syndrome (MODS). Not only does severe burn triggers secondary pathology, but even relatively minor burn injuries can also cause adverse systemic effects [23,95]. A prolonged existence of SIRS can lead to severe muscle protein catabolism, described as persistent inflammation, immunosuppression, and catabolism syndrome, and is associated with an increased risk of multi-organ failure and death [93,96,97]. 

#### 4.1.1. Altered Endogenous Steroid Biosynthesis after Burn

Burn injury is followed by a persistent hypermetabolic response for up to two years after the burn, resulting in changes in the endogenous production of besides inflammatory mediators, steroids as well [98,99]. Plasma catecholamines (i.e., adrenalin, noradrenalin and dopamine), glucagon, and cortisol can be elevated by up to 50-fold, which leads to whole-body catabolism, elevated resting energy expenditures and multiorgan dysfunction [98]. Moreover, gender-related differences in the endogenous production of adrenal and gonadal steroids have been reported [100]. Decreased testosterone concentrations and elevated oestrone concentrations were found up to 21 days post-burn. In addition, glucocorticoids’, progestogens’, and androgen precursors’ concentrations positively correlated with the % TBSA burned [100].

Interventions in order to modulate (somewhat) the profound hypermetabolic response after burn, which resulted in significantly decreased morbidity, include, e.g., early excision and grafting of burns, thermoregulation, control of infection, early and continuous enteral nutrition and pharmacologic treatments [98].

#### 4.1.2. Haemodynamic Failure

The first hours post-burn form a critical phase in which several serious systemic complications can develop in the burn patient, with the risk of mortality. In the first place, there is an acute danger of developing a shock state after severe burn trauma, which could develop within hours post-burn [101]. Patients reach a hypodynamic state directly after burn trauma, which is characterised by increased vascular permeability as a result of high thermal energy and an increase in, amongst others, pro-inflammatory mediators, nitric oxide and prostaglandins. This increase in vascular permeability could cause massive volume depletion via the extravasation of blood plasma to the interstitial tissue. The extravasated plasma proteins lead to an altered osmotic gradient, which further enhances tissue oedema and loss of blood volume [102]. However, mortality due to haemodynamic failure in the acute phase post-burn is rare since clinical interventions have greatly improved over time [103,104]. However, serious morbidity could develop in the early phase post-burn when fluid resuscitation is inappropriate. Adequate and large-volume resuscitation is required to preserve adequate perfusion of all organs, specifically the kidneys, to prevent early acute kidney injury. Also, fluid overload can be detrimental and compromise end-organ function [96,101]. In a multicentre study, it was found that when fluid resuscitation exceeded the calculated needed volume by 25%, there was a higher chance of mortality [105]. These findings illustrate that adequate fluid resuscitation is of great importance in the outcome post-burn.

#### 4.1.3. Sepsis

Burn-induced disruption of the skin barrier considerably increases the risk of infection, which could lead to sepsis post-burn. Septic patients have a systemic infection to which the immune system strongly reacts, and this overwhelming response could affect multiple organs [106]. Moreover, sepsis could further enhance the immunosuppression that is observed after burn injury since this systemic disease induces apoptosis of immune cells [107]. Thus, septic patients are in a hyperinflammatory state leading to an immunosuppressive state [108]. When burn patients develop sepsis, this is usually observed several days after injury, at the time that circulatory problems often have stabilised [104]. Factors that further contribute to the occurrence of sepsis are likely related to the immune response post-burn. For example, the hyperactive macrophage phenotype seems to have an important role in the development of sepsis by the release of increased amounts of inflammatory cytokines. High concentrations of these inflammatory mediators could subsequently lead to alterations in the immune system, such as the lymphocyte response, with an impaired immune function as an outcome [42]. As such, higher levels of the anti-inflammatory cytokine IL-10, were found in septic burn patients than in non-septic burn patients [30]. 

It is generally known that early diagnosis of sepsis in a burn patient, with (still) negative blood cultures, is difficult since severe burn injury typically results in a hypermetabolic and highly inflammatory state. Therefore, burn patients often present with sepsis-related criteria, irrespective of systemic infection [33,98,109,110,111]. 

Still, a better prediction system for early sepsis diagnosis in burn patients could enable improved, earlier treatment of the septic burn patient. Biomarkers might give additional predictive value for sepsis in burn patients, such as certain systemic or local cytokine levels [112,113,114]. In this respect, procalcitonin (PCT), a 116-amino acid polypeptide prohormone of calcitonin, has become an important biomarker to aid in the diagnosis of bacterial sepsis. It has a high potential to improve the clinical assessment of patients [115]. In a normal situation, a very low concentration of procalcitonin is present in the blood. However, production can be stimulated in almost every organ by inflammatory cytokines and especially bacterial endotoxins, for example, in sepsis, which causes large amounts of PCT to be released into the blood. As a result, the amount of PCT could be seen as a potential biomarker of, for example, sepsis. The higher the PCT concentration, the more likely systemic infection and sepsis would be [116]. However, it is very difficult to accurately diagnose sepsis based on biomarkers only [117]; even positive blood cultures in sepsis cohorts are found in around 40% in prospective studies [118,119]. Moreover, sepsis diagnosis in burn patients is based on the clinical situation, i.e., increased fluid requirements, low platelet counts and declining pulmonary and/or renal function.

#### 4.1.4. Acute Impact on Multiple Organ Systems

Severe burn trauma could damage multiple remote organs in the acute phase, which in the end, can lead to the loss of organ function (Figure 4A). This multiple organ dysfunction syndrome (MODS) is often observed in combination with sepsis in burn patients; however, the enormous inflammatory response to burn-induced tissue damage could also lead to MODS without sepsis, correlated to both the burn wound size and depth of burn [120,121]. 

One of the most often affected organs in the acute phase post-burn is the kidneys [120,121,125]. Early acute kidney injury occurs in the first days post-burn and is mostly due to a combination of hypovolemia, cardiac dysfunction and denatured proteins that exert a toxic effect on the kidneys. Multiple factors contribute to late acute kidney injury as well; among these is inflammation-induced damage to renal tubular epithelium and renal arteries [125]. Next to the kidneys, the respiratory system is often involved in MODS. Burn patients are at risk of developing acute respiratory distress syndrome (ARDS), which is characterised by pulmonary fluid infiltrates and hypoxemia [122,126]. Some important factors that have been shown to correlate with the development of ARDS post-burn are larger burn wound size and larger full-thickness burn wound area, higher age, too-aggressive fluid resuscitation, and the presence of pneumonia or acute kidney injury [127]. The exact pathophysiology of ARDS is unknown, but largely contributing is the burn-induced inflammatory response that is characterised by infiltrating neutrophils and increased concentrations of pro-inflammatory cytokines. This inflammatory response leads to damage to the endothelium of lung microvasculature and the epithelium of alveoli, resulting in the extravasation of fluid from the vasculature into the alveoli [126]. In addition, acute kidney injury likely enhances respiratory inflammation post-burn [128]. 

In addition, often observed in the early phase post-burn are haematologic alterations. In short, burn trauma causes alterations in the balance between pro- and anticoagulant factors and damage to the endothelium of the local and systemic vasculature. This could lead to a hypercoagulable state in the burn patient [123,124]. Also, changes in platelet concentrations post-burn, namely early thrombocytopenia, followed by thrombocytosis, could contribute to coagulopathy (Figure 4B) [129,130]. Clinically, these systemic changes could have a major impact on the patient’s prognosis. For example, low platelet concentrations might enhance the risk of bleeding during surgical interventions post-burn. In contrast, the hypercoagulable state of burn patients could increase the risk of deep venous thrombosis (DVT) or pulmonary embolism. 

Other organ systems that could be affected in the early phase post-burn include the liver, the heart, the gastrointestinal tract and the central nervous system. In general, contributing to early post-burn dysfunction of these organ systems are the hypodynamic state and factors that stimulate oedema formation in organs, such as inflammation and fluid resuscitation [98]. For example, the impaired production of constitutional liver proteins often occurs as a consequence of liver oedema upon burn trauma. Although the precise cause is unknown, burn patients often suffer from increased abdominal pressure as well, which could result not only in hypoperfusion of the visceral organs but also in cardiac and respiratory failure [131]. Furthermore, activation of the stress response directly impacts the physiology of the heart by increasing, amongst others, the heart rate, ejection fraction and oxygen needs [132]. However, the latter, in turn, causes cardiac depression due to high cytokine release.

#### 4.1.5. Long-Term Impact on Multiple Organ Systems

In the years following burn trauma, it is hypothesised that the systemic immune response to burn frequently has a negative impact on multiple organ systems. Recent studies show that patients with a history of burn trauma are potentially at increased risk of developing cardiovascular diseases, metabolic syndrome, diabetes, musculoskeletal problems, infectious diseases and cancer [50,133]. For example, several studies have shown a higher rate of hospital admissions for cardiovascular disease among burn patients [134,135,136]. From a pathophysiological point of view, increased concentrations of inflammatory mediators, including IL-1β, IL-6 and TNF-α, result in, amongst others, decreased cardiac contractility [137,138], which may be the biological mechanism through which increased cardiovascular disease risk may arise. 

Loss of muscle strength and endurance post-burn also has a major impact on burn patients. Years after burn trauma, patients have reported fatigue, swelling and pain in joints and weakness of the limbs. Biological mechanisms that underlie the loss of skeletal muscle mass post-burn include the effects of the stress hormone catecholamine, burn-induced insulin resistance, and increased need for amino acids for wound healing and the immune system. Also, inflammatory cytokines mediate a catabolic state post-burn [139]. 

Besides the loss of muscle mass, post-burn changes in endocrine and metabolic pathways can induce alterations in bone tissue. Remodelling of bone tissue is a normal physiological process. However, changes in the activity of the bone-resorbing osteoclasts and the mineralising osteocytes in burn patients could lead to decreased bone mineral density and bone mass. Involved in these changes in bone remodelling are, amongst others, the adrenal stress hormone glucocorticoid and the cytokines IL-1β and IL-6 [140].

### 4.2. Prediction of the Clinical Outcome

#### 4.2.1. The Cytokine Network as Clinical Predictor

Burn injury elicits a massive inflammatory response that results in detectable plasma levels of inflammatory cytokines, which typically peak in concentration within the first week post-burn. Several research groups have questioned whether the blood concentrations of cytokines or other inflammatory proteins might be valuable as predictors of clinical outcomes post-burn [29,32,36,113,141]. As such, cytokine profiles were identified that correlated with outcomes in the first month after injury in adult burn patients with mild to severe injuries [36]. On the day of burn trauma, plasma levels of IL-6, IL-8, IL-10 and CCL2 were significantly higher in non-survivors than in survivors. Furthermore, this cytokine profile of the early phase post-burn correlated with Sequential Organ Failure Assessment (SOFA) scores, a scoring system that often is used to determine the severity of MODS which is based on the functionality of six vital organ systems (i.e., the respiratory, haematologic, hepatic, cardiovascular, neurologic and renal system). Although IL-6, IL-8, IL-10 and monocyte chemoattractant protein 1 (MCP-1) levels correlated to the size of injury 24–48 h post-burn, these cytokines seemed not very dependable clinical markers of outcome compared to demographic data, e.g., age, size of the injury and inhalation injury, with the exception of IL-8 and MCP-1 levels on admission in predicting death [29]. Besides, metalloproteinases (MMPs) were also studied for their association with outcome. Although MMP-8 and -9 were higher in burn patients than in healthy controls, they did not correlate with % TBSA and were not associated with clinical severity or outcome measures. TIMP-1, in contrast, which was also higher in patients than in healthy controls, was independently associated with 90-day mortality, correlated with % TBSA burned, fluid and noradrenaline requirement and SOFA score [142]. Therefore TIMP-1 may serve as a potential biomarker in the outcome of burn patients. However, further research is necessary in order to reveal the biological background for the outcome association.

Overall, several studies point to the importance of cytokines as indicators for the severity of the disease. However, due to the integrative character of the post-burn systemic immune response, with the involvement of many inflammatory factors, it seems impossible to attribute individual cytokine levels to a specific patient prognosis. Therefore it is necessary to use cytokine profiles.

#### 4.2.2. Molecular Markers as Clinical Predictors

In addition to markers of the post-burn inflammatory response, other markers, including molecular markers, e.g., genomic DNA markers, mRNA/miRNAs/lncRNAs/circRNAs markers, epigenetic markers, proteomics, and metabolics, could serve as potential biomarkers in the outcome of burn patients [143]. The advantage of these non-inflammatory response markers could be that they can predict, for instance, sepsis in burn patients, as markers of the post-burn inflammatory response do not reflect the severity of the infection. 

Analyses of changes in genetic processes in the skin during burns revealed three potential novel diagnostic markers in blood from burn patients. Three Hub genes, MCEMP1, MMP9, and S100A12, were shown to be significantly elevated in burn patients and were suggested as key blood biomarkers that can be used to identify skin damage in burn patients [144].

Furthermore, analyses of miRNAs that are potentially involved in early burn-response gene regulation have shown that miRNA-497-5p, which regulates skin cell regeneration, was downregulated in tissue and dermal interstitial fluid (dISF) of burned skin. Therefore, further examination of miRNA-497-5p as a biomarker for the severity of burns is suggested [145].

## 5. Conclusions

In conclusion, burn wounds clearly differ from other types of skin trauma or wounds in their extensiveness in width and depth, in their substantial risk of deepening to a more necrotic wound, and in the high incidence of pathological scar formation. The main conclusion of this review is that burn wound healing largely depends on the inflammatory environment. However, burn wounds cause complex local and systemic pathology, both in the early phase and in the long-term. For a long time, the focus of burn research was on the acute phase. To date, early burn wound excision, infection prevention and appropriate fluid resuscitation can often prevent or limit a (septic) shock after a severe burn injury. However, it becomes increasingly apparent that burn injury adversely affects systemic physiology in the long-term, such as metabolism and heart functionality, and that also milder burn injury could be detrimental beyond the site of the initial injury. The local and systemic post-burn disease occurs mainly due to an excessive and prolonged inflammatory response, which makes burn trauma distinct from other forms of trauma, but on the other hand, it has many similarities with, among others, severe sepsis [97]. Although many in vitro, in vivo, and clinical studies have broadened the knowledge on post-burn immunology, this review identifies the gap in knowledge where the area of burn research requires a better understanding of how different immune factors and pathways are involved in the persistence of inflammation and how they link with local and systemic pathology. A challenging task for further burn research is to integrate the variety of data on involved immunological factors and to develop research models that accurately represent human physiology. This could be advanced in vitro models, such as skin-on-chip models [146,147] that represent all skin layers and adnexal structures, or in silico models [148]. An in silico systems biology approach to burn immunology could be of great value since computer models seem better able to organise and integrate complex information than the human mind. A systems biology approach to human burn physiology, in combination with advanced human in vitro models and clinical studies, could ultimately lead to new scientific immunological insights and could help to improve the care of burn patients [149,150].

## Figures and Tables

**Figure 1 cells-12-00345-f001:**
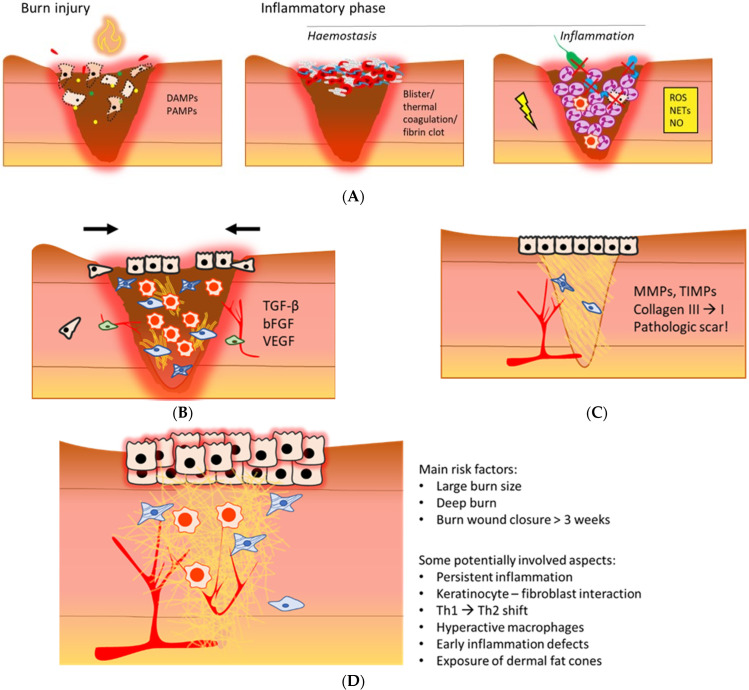
An illustrative overview of the post-burn inflammatory phase (**A**), proliferation phase (**B**), remodelling phase (**C**), and a hypertrophic scar (**D**) [6,7,8,9,10]. (**A**) Necrotic tissue and infiltrating pathogens post-burn initiate an inflammatory response that is mediated by local increases in DAMPs and PAMPs. Depending on burn depth, haemostasis is maintained via blister formation, thermal coagulation of dermal blood vessels, or activation of the coagulation cascade. During inflammation, neutrophils, the complement system and macrophages are main contributors to pathogen elimination and necrotic tissue clearance, mainly via phagocytosis and oxidative stress. Persistent inflammation could further damage the wound bed. DAMPs: damage-associated molecular patterns; PAMPs: pathogen-associated molecular patterns; ROS: reactive oxygen species; NETs: neutrophil extracellular traps; NO: nitric oxide. (**B**) The anti-inflammatory macrophage is a dominant immune cell in this phase. Proliferation is characterised by formation of granulation tissue, reepithelisation, angiogenesis and wound contraction, mediated by a variety of cytokines and growth factors. TGF-β: transforming growth factor β; bFGF: basis fibroblast growth factor; VEGF: vascular endothelial growth factor. (**C**) Remodelling increases scar tissue strength, and important proteins in this phase are MMPs and TIMPs. The outcome of the local wound healing phases post-burn often is a pathological scar. MMPs: matrix metalloproteinases; TIMPs: tissue inhibitors of matrix metalloproteinases. (**D**) A hypertrophic scar is red, rigid and raised. A typical aspect is excessive ECM deposition. The main risk factors of HTS formation and potentially involved causal aspects are summarised on the right.

**Figure 2 cells-12-00345-f002:**
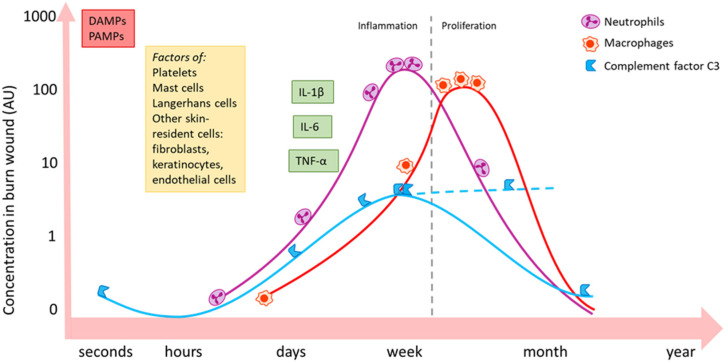
Schematic summary of the local activity of some important inflammatory mediators and immune cell types post-burn [10,25,26]. Burn induces an immediate increase in local concentration of DAMPs and PAMPs via necrotic tissue and infiltrating pathogens. Skin-resident (immune) cells become active and release inflammatory mediators and could change, for example, their expression of certain receptors. Neutrophils (purple) massively migrate into the wound bed during the inflammatory phase and follow a similar pattern as complement factor C3 (blue). C3 levels might continuously increase in later phases (dashed line). Macrophages (red) peak in the later proliferation phase. The influx of immune cells is paralleled by an increase in inflammatory cytokines and growth factors, such as IL-1β, IL-6 and TNF-α. Local persistence of inflammation might continue up to weeks post-burn. DAMPs: damage-associated molecular patterns; PAMPs: pathogen-associated molecular patterns; IL: interleukin; TNF: tumour necrosis factor.

**Figure 3 cells-12-00345-f003:**
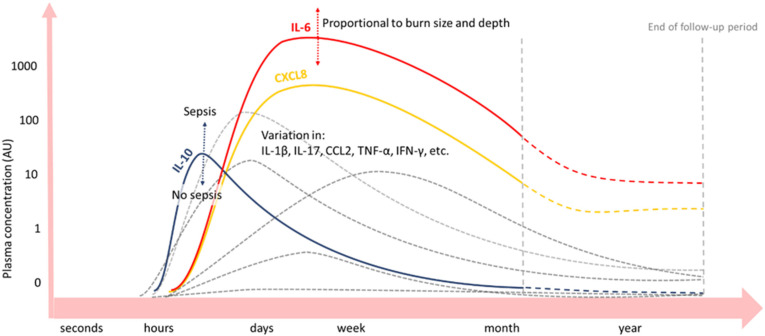
Blood concentrations of cytokines and growth factors after severe burn injury [29,30,31,32,33,34,35,36]. Inflammatory cytokines IL-6 (red) and CXCL8 (yellow) are significantly increased up to a month or even years (dashed lines) post-burn. IL-10 (blue) peaks on day one post-burn and declines hereafter. Peak concentrations of IL-6 and IL-10 depend on burn wound size and depth and presence of sepsis, respectively. Plasma concentrations over time of other inflammatory mediators vary among burn patient follow-up studies (grey lines).

**Figure 4 cells-12-00345-f004:**
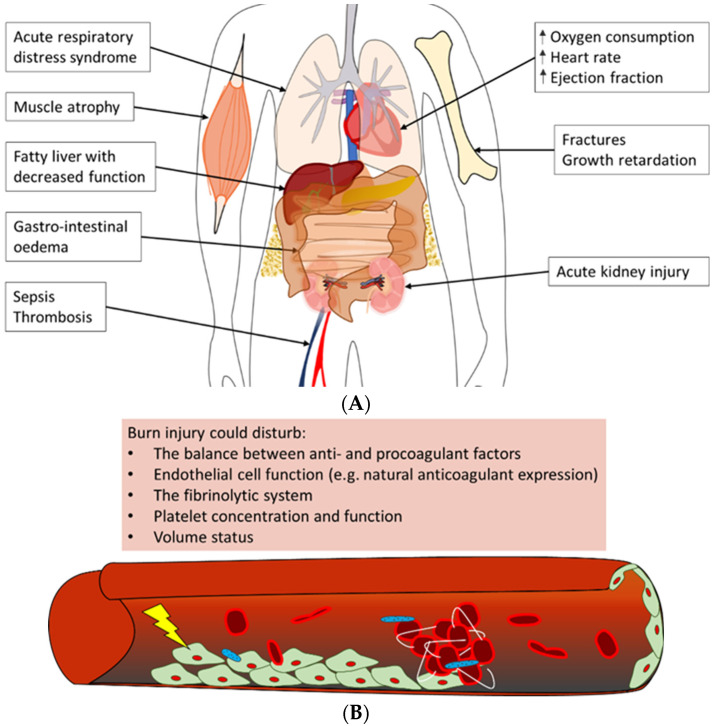
Overview of often observed systemic complications of burn injury [106,121,122,123,124]. (**A**) Burn patients could develop a variety of complications post-burn. Summarised are possible adverse outcomes of (severe) burn injury in the acute phase, e.g., acute respiratory distress syndrome and acute kidney injury, as well as long-term negative effects, such as growth retardation and a fatty liver. (**B**) The main trigger of coagulopathy is the inflammatory response post-burn. This response could lead to disturbances in factors that contribute to haemostasis, e.g., pro- and anticoagulant factors, platelet concentration and endothelial cell function, which could result in thrombotic events in burn patients.

**Table 1 cells-12-00345-t001:** An overview of the local events after burn based on the major “actors” and their actions.

Actor	Action	References
Damaged skin cells	become necrotic ⇨ secrete damage-associated molecular patterns (DAMPs)	[7,8,9]
Pathogens	break through skin barrier ⇨ secrete pathogen-associated molecular patterns (PAMPs)	[6]
Skin-resident immune cells (mast cells, Langerhans cells)	become activated ⇨ release inflammatory cytokines (e.g., histamine, bradykinin, cathepsins and proteases) ⇨ vascular permeability; and activate other immune cells (i.e., neutrophils, macrophages)	[26]
Keratinocytes, fibroblasts and endothelial cells	release of cytokines and growth factors that attract and activate immune cells	[25]
Epithelial cells, endothelial cells and macrophages	release of interleukin 8 (IL-8), a.k.a. CXCL8, ⇨ recruitment of neutrophils	[25]
Neutrophils	phagocytosis of opsonized pathogens, release of reactive oxygen species (ROS), and formation of neutrophil extracellular traps (NETs)	[25]
	release of IL-1 and tumor necrosis factor α (TNF-α) ⇨ recruitment monocytes from the blood	
Keratinocytes, fibroblasts, endothelial cells	release of CCL2 (a.k.a. MCP-1) ⇨ recruitment of macrophages (arise from blood monocytes); dendritic cells, natural killer cells and lymphocytes	[25]
M1 macrophages (pro-inflammatory)	removal of tissue debris and pathogens via phagocytosis; release inflammatory cytokines (e.g., IL-6, IL-1β and TNF-α)	[10]
M2 macrophages (tissue repair stimulating)	important in proliferation and remodeling phase of wound healing	[10]
γδ T cells	release of IFN-γ ⇨ induces a hyperactive macrophage phenotype	[28]
Hyperactive macrophages	release of increased amounts of the pro-inflammatory cytokines IL-1 and IL-6, and of transforming growth factor β (TGF-β)	[28]

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
