# Peer review of "The Complexity of the Post-Burn Immune Response: An Overview of the Associated Local and Systemic Complications"

_cells, 2023, doi:10.3390/cells12030345_

Round 1

Reviewer 1 Report

This review is very interesting and certainly can add insight for the scientific community. There is a lot of information in the text and my only concern is that it is not organised in a way that makes the information easy to find or understand. I believe the major questions regarding the post burn immune response are how long the different phases last and how can we identify them, what is proinflammatory and what is antiinflammatory, what are the clinical implications/how can this information aid treatment? I would suggest to arrange the text under subheadings like innate and adaptive response (and cells and mediators under) OR under pro- vs antiinflammatory response. 

Another important aspect of burn care is diagnosis sepsis in burn injury which the authors have already identified is not an easy task. The authors suggest PCT but I suggest to also add that the use of PCT can be questioned (Tang 2007). To date, no biomarkers can accurately diagnose sepsis, and even positive blood cultures in sepsis cohorts are found in around 40% in well conducted and prospective studies (Nannan 2019). The sepsis diagnosis in burn injury patients is based on clinical grounds, usually based on increased fluid requirements, low platelet counts and declining pulmonary/renal function. 

Another important perspective if the authors intend to describe the immune response after burn is the endogenous steroids and how they are altered following the injury and inflammation, especially the endogenous cortisol following HPA axis perturbations.

The authors state that IL-6 and IL-8 have been shown to correlate with the size of the burn, which I agree with, but interestingly much of the immune response does not correlate with the size of the burn, ie even a small injury still gives rise to a major immune response. Can the authors comment on why this is?

Which of these different cell types, cytokines and other mediators have been found in blood (most) and in tissue (not so much data) and what are the gaps here?

The authors discuss the use of cytokines as predictors of outcome in 4.2 and in fact cytokines are not very dependable clinical markers of outcome compared to demographic data such as patient age, size of the injury and inhalation injury. In ref 25 MCP-1 (in addition to IL-8, already mentioned in the review) is introduced as a potential marker for outcome and I cannot find this discussed in the review. Other gaps may be metalloproteases which are not mentioned such as MMPs and TIMP, described by Hastbacka 2015.

What are the authors hoping that future studies will answer and what information would aid the treatment of the patient in the different phases following injury. 

Finally, I cannot find the the statements about mast cells on page 2 under heading 2 to be covered by reference 14. 

Author Response

Point by point reply

First of all we would like to thank the reviewers for evaluating our work, which has given us the opportunity to improve our manuscript. We have read their comments carefully and have altered our manuscript accordingly. All changes to the text in the manuscript are made using the “Track Changes” function in the revised manuscript and are pointed out below.

Reviewer comments

Reviewer #1:

Comments and Suggestions for Authors:

  1. This review is very interesting and certainly can add insight for the scientific community. There is a lot of information in the text and my only concern is that it is not organised in a way that makes the information easy to find or understand. I believe the major questions regarding the post burn immune response are how long the different phases last and how can we identify them, what is proinflammatory and what is antiinflammatory, what are the clinical implications/how can this information aid treatment? I would suggest to arrange the text under subheadings like innate and adaptive response (and cells and mediators under) OR under pro- vs antiinflammatory response.

Answer:               

We agree with the reviewer and arranged the text in “2. The post-burn immune response” on page 2-4  under the following subheadings: 

  1. The post-burn immune response

2.1. The innate immune response after burn

2.1.1 Acute phase response (APR) after burn

2.1.2. A prolonged and over-active systemic APR after burn injury: a pivotal role for complement

2.2. The adaptive immune response after burn

We did not choose for subheadings like pro- vs anti-inflammatory response, since both subgroups are involved in different phases, which would make it even more confusing.

  1. Another important aspect of burn care is diagnosis sepsis in burn injury which the authors have already identified is not an easy task. The authors suggest PCT but I suggest to also add that the use of PCT can be questioned (Tang 2007). To date, no biomarkers can accurately diagnose sepsis, and even positive blood cultures in sepsis cohorts are found in around 40% in well conducted and prospective studies (Nannan 2019). The sepsis diagnosis in burn injury patients is based on clinical grounds, usually based on increased fluid requirements, low platelet counts and declining pulmonary/renal function.

Answer:

We agree with the reviewer and would like to thank the reviewer for this valuable addition. We adjusted the text (and added relevant references) on page 9 of the revised manuscript accordingly to:

“As a result, the amount of PCT can be used could be seen as a potential biomarker of, for example, sepsis. The higher the PCT concentration, the more likely systemic infection and sepsis is would be [116]. However, it is very difficult to accurately diagnose sepsis based on biomarkers only [117], even positive blood cultures in sepsis cohorts are found in around 40% in prospective studies [118,119]. Moreover, sepsis diagnosis in burn patients is based on clinical situation e.g. i.e. increased fluid requirements, low platelet counts and declining pulmonary and/or renal function.”

  1. Another important perspective if the authors intend to describe the immune response after burn is the endogenous steroids and how they are altered following the injury and inflammation, especially the endogenous cortisol following HPA axis perturbations.

Answer:

We agree including the alteration of the endogenous steroids after burn will be of added value for the description of the (systemic) immune response after burn.

We added the following paragraph on page 8 of the revised manuscript:

“ 4.1.1 Altered endogeneous steroid biosynthesis after burn

Burn injury is followed by persistent hypermetabolic response for up to two years after burn, resulting in changes in the endogenous production of besides inflammatory mediators, steroids as well [98, 99]. Plasma catecholamines (i.e. adrenalin, noradrenalin and dopamine), glucagon, cortisol can be elevated by up to 50-fold, which lead to whole-body catabolism, elevated resting energy expenditures and multiorgan dysfunction [98]. More-over, gender-related differences in the endogenous production of adrenal and gonadal steroids have been reported [100]. Decreased testosterone concentrations and elevated estrone concentrations were found up to 21 days post-burn. In addition, glucocorticoids’, progestogens’ and androgen precursors’ concentrations positively correlated with % TBSA burned [100].

Interventions in order to modulate (somewhat) the profound hypermetabolic response after burn, which resulted in significant decreased morbidity, include e.g. early excision and grafting of burns, thermoregulation, control of infection, early and continuous enteral nu-triton and pharmacologic treatments [98].”

  1. The authors state that IL-6 and IL-8 have been shown to correlate with the size of the burn, which I agree with, but interestingly much of the immune response does not correlate with the size of the burn, ie even a small injury still gives rise to a major immune response. Can the authors comment on why this is?

Answer:

Indeed, we and others have shown that even small burns can give rise to major immune response, also systemic effects. However, we think that not only the size, but in addition, and maybe more important, the depth of a burn wound matters. For instance in previous studies we have shown that even burn wound with 1 – 2 % TBSA burned, showed systemic inflammation, however, these were full thickness burn wounds (3d degree) [1, 2].  

This does not directly explain why some of the immune mediators do and some of them do not correlate with the size of the burn, but it could be that for the very early (acute) stage inflammation factors like IL-6, IL-8 and CRP a small and relatively superficial wound is already enough to make them increase, but for others the depth of the wound matters more. 

  1. Korkmaz HI, Ulrich MMW, van Wieringen WN, Vlig M, Emmens RW, Meyer KW, Sinnige P, Krijnen PAJ, van Zuijlen PPM, Niessen HWM. The Local and Systemic Inflammatory Response in a Pig Burn Wound Model With a Pivotal Role for Complement. J Burn Care Res. 2017 Sep/Oct;38(5):e796-e806.
  2. Korkmaz HI, Ulrich MMW, Vogels S, de Wit T, van Zuijlen PPM, Krijnen PAJ, Niessen HWM. Neutrophil extracellular traps coincide with a pro-coagulant status of microcirculatory endothelium in burn wounds. Wound Repair Regen. 2017 Aug;25(4):609-617.

  1. Which of these different cell types, cytokines and other mediators have been found in blood (most) and in tissue (not so much data) and what are the gaps here?

Answer:

For this, we would like to refer to figure 2 “Schematic summary of the local activity of some important inflammatory mediators and immune cell types post-burn [22 ,68, 75].” and figure 3 “Blood concentrations of cytokines and growth factors after severe burn injury [25-32].” on page 16 of the manuscript. Here we summarized the different cell types, cytokines and other mediators that have been found in blood and in tissue.

According to this, the major gap is actually data from both blood and tissue from the same (cohort of) patients. This would give a better and more complete overview and may also enable i.e. correlation analyses between blood and tissue levels.

  1. The authors discuss the use of cytokines as predictors of outcome in 4.2 and in fact cytokines are not very dependable clinical markers of outcome compared to demographic data such as patient age, size of the injury and inhalation injury. In ref 25 MCP-1 (in addition to IL-8, already mentioned in the review) is introduced as a potential marker for outcome and I cannot find this discussed in the review. Other gaps may be metalloproteases which are not mentioned such as MMPs and TIMP, described by Hastbacka 2015.

Answer:

We agree including these potential markers  will be of added value to 4.2.

We adjusted the text accordingly on page 11 of the revised manuscript:

“ 4.2.1 The cytokine network as clinical predictor

Burn injury elicits a massive inflammatory response that results in detectable plas-ma levels of inflammatory cytokines, which typically peak in concentration within the first week post-burn. Several research groups have questioned whether the blood concentrations of cytokines or other inflammatory proteins might be valuable as predictors of clinical outcome post-burn [25, 28, 32, 111113, 136141]. As such, cytokine profiles  where identified that correlated with outcome in the first month after injury in adult burn patients with mild to severe injuries [32]. At the day of burn trauma, plasma levels of IL-6, IL-8, IL-10 and CCL2 were significantly higher in non-survivors than in survivors. Furthermore, this cytokine profile of the early phase post-burn correlated with Sequential Organ Failure Assessment (SOFA) scores, a score system that often is used to determine the severity of MODS which is based on the functionality of six vital organ systems (i.e. the respiratory, hematologic, hepatic, cardiovascular, neurologic and renal system). Although IL-6, IL-8, IL-10 and monocyte chemoattractant protein 1 (MCP-1) levels correlated to the size of injury 24 – 48 hours post-burn, these cytokines seemed not very dependable clinical markers of outcome compared to demographic data e.g. age, size of the injury and inhalation injury, with the exception of IL-8 and MCP-1 levels on admission in predicting death [25]. Besides, metalloproteinases (MMPs) were also studied for their association with out-come. Although, MMP-8 and -9 were higher in burn patients than in healthy controls, they did not correlate with % TBSA and were not associated with clinical severity or outcome measures. TIMP-1 in contrast, which was also higher in patients than in healthy controls, was independently associated with 90-day mortality, correlated with %TBSA burned, flu-id and noradrenaline requirement and SOFA score [142]. Therefore TIMP-1 may serve as a potential biomarker in outcome of burn patients. However, further research is necessary in order to reveal he biological background for the outcome association.

Overall, several studies point to the importance of cytokines as indicators for severity of disease. However, due to the integrative character of the post-burn systemic immune response, with involvement of many inflammatory factors, it seems impossible to attribute individual cytokine levels to a specific patient prognosis. Therefore it is necessary to use cytokine profiles.”

  1. What are the authors hoping that future studies will answer and what information would aid the treatment of the patient in the different phases following injury.

Answer:

We hope that future studies will answer or at least give more insight in how different immune factors and pathways are involved in the persistence of inflammation and how they link with local and systemic pathology. The latter in particular would be of added value for the treatment of patients. In addition, knowledge about “timing” of intervention would aid the treatment of patients in the different phases post-burn.

As we stated in the last sentence of our manuscript: “A systems biology approach to human burn physiology, in combination with advanced human in vitro models and clinical studies, could ultimately lead to new scientific immunological insights, and could help to improve the care of burn patients”.

  1. Finally, I cannot find the statements about mast cells on page 2 under heading 2 to be covered by reference 14.

Answer:

It was actually meant that the statements about mast cells were covered by the reference “Douaiher, J., et al., Development of mast cells and importance of their tryptase and chymase serine proteases in inflammation and wound healing. Adv Immunol, 2014.”; apparently there went something wrong, our apologies for this.

Reference 14 has been replaced in the revised manuscript.

Reviewer 2 Report

This submitted review reports the complexity of post-burn immunology, focusing on both the local and systemic complications. It is a worthy topic to understand and treat the exaggerated and prolonged inflammation after burns. However, it would only be possible to recommend the publication of this manuscript after incorporating these major changes:

1. Inside the body, the innate immune system consists of macrophages, neutrophils, DCs, NK cells, mast cells, and more. The roles of all these innate immune cells during burn injury should be described in detail and current references should be provided.

2. In “Local and systemic complications related to a dysregulated immune response”, the dysregulated characteristics of immune cells should be discussed.

3. In “Prediction of the clinical outcome”, besides cytokine networks, miRNA and others may be discussed.

Author Response

Point by point reply

First of all we would like to thank the reviewers for evaluating our work, which has given us the opportunity to improve our manuscript. We have read their comments carefully and have altered our manuscript accordingly. All changes to the text in the manuscript are made using the “Track Changes” function in the revised manuscript and are pointed out below.

Reviewer comments

Reviewer #2:

Comments and Suggestions for Authors:

This submitted review reports the complexity of post-burn immunology, focusing on both the local and systemic complications. It is a worthy topic to understand and treat the exaggerated and prolonged inflammation after burns. However, it would only be possible to recommend the publication of this manuscript after incorporating these major changes:

  1. Inside the body, the innate immune system consists of macrophages, neutrophils, DCs, NK cells, mast cells, and more. The roles of all these innate immune cells during burn injury should be described in detail and current references should be provided.

Answer:               

Regarding the roles of macrophages, neutrophils, DCs, NK cells, mast cells and other immune cells during the innate immune system; the role of these cells in the innate immune system has already been described in many other reviews about the immune response and/or wound healing in general. Therefore we chose not to repeat this (as that would not bring anything news) and to focus on the role of immune cells and factors that are currently specifically known and described in burns.

  1. In “Local and systemic complications related to a dysregulated immune response”, the dysregulated characteristics of immune cells should be discussed.

Answer:

Regarding the dysregulated characteristics of immune cells in local and systemic complications, first of all we would like to emphasize that the characteristic of immune cells that are dysregulated is not fully understood yet. The main characteristic that applies to most immune cells (and factors) is that they persist locally in the burn wound and remain elevated systemically in blood for a long time, compared to non-burn wounds.

The following characteristics had been discussed (highlighted in yellow in the revised manuscript):

Local:

  • Page 5: “The inflammatory response is essential for wound healing, however, persistent inflammation could result in increased local damage, eventually leading to burn wound deepening. A relation exists between local complement persistence and presence of inflammatory cell infiltrates [9, 20]. This suggests that the complement cascade contributes to excessive inflammation, thereby mediating burn wound deepening.”
  • Page 5: “Neutrophils, as one of the local infiltrating immune cells, could cause local damage to the microvasculature and surrounding tissue, via release of pro-inflammatory cytokines [56] and probably via NETs [55].”
  • Page 6: “For example, high concentrations of TGF-β could prevent myofibroblasts apoptosis and induce myofibroblasts formation/differentiation, resulting in uncontrolled wound con-traction and excessive production of ECM [66].”
  • Page 6: “Also, the degree of (myo)fibroblast activity in burn patients depends on which Th cell phenotype dominates the environment. Th1 cells mainly release IFN-γ and IL-12, thereby downregulating (myo)fibroblast activity. In contrast, Th2 cells primary produce cytokines that stimulate (myo)fibroblast activity, namely the cytokines IL-4, IL-5, IL-10 and IL-13 [3, 67].”
  • Page 6: “Another paracrine actor is the mast cell, which could also modulate activity of (myo)fibroblasts [68]. Although the exact mediator(s) is not known, several studies have shown that mast cell proteases have fibrinogenic effects. E.g. tryptase stimulates procollagen mRNA synthesis, contraction and differentiation of dermal fibroblasts into myofibroblasts. Histamine, another mast cell mediator, can also increase collagen production in fibroblasts [69, 70].”
  • Page 6-7: “Since platelets release the fibroblast stimulating platelet-derived growth factor (PDGF), overactivity of platelets could lead to hypertrophic scarHTS formation [74].”
  • Page 7: “Deep dermal fibroblasts produce more ECM, more TGF-B and CTGF and less MMPs than the more superficial fibroblasts [82, 89], thereby promoting excessive scarring”.

Systemic:

  • Page 7: “Severe burn injuries induce a systemic response that is described as the systemic inflammatory response syndrome (SIRS) is caused by an “over-exuberant” acute phase inflammation [91-94].”
  • Page 8: “For example, the hyperactive macrophage phenotype, seems to have an important role in the development of sepsis by the release of increased amounts of inflammatory cytokines. High concentrations of these inflammatory mediators could subsequently lead to alterations in the immune system, such as the lymphocyte response, with an impaired immune function as outcome [38].”
  • Page 9: “The exact pathophysiology of ARDS is unknown, but largely contributing is the burn-induced inflammatory response that is characterized by infiltrating neutrophils and increased concentrations of pro-inflammatory cytokines. This inflammatory response leads to damage to endothelium of lung microvasculature and to epithelium of alveoli, resulting in extravasation of fluid from the vasculature into the alveoli [120].
  • Page 9: “Also, changes in platelet concentrations post-burn, namely an early thrombocytopenia, followed by thrombocytosis, could contribute to coagulopathy (figure 4B) [126, 127].”

  1. In “Prediction of the clinical outcome”, besides cytokine networks, miRNA and others may be discussed.

Answer:

We initially did not discuss other non-inflammatory response markers, as this review focusses on the post-burn immune response and the main inflammatory markers. Nevertheless, we agree with the reviewer that this is a valuable addition.

We included the following subheading and text on page 11 – 12 of the revised manuscript:

“4.2.2 Molecular markers as clinical predictor

In addition to markers of the post-burn inflammatory response other markers, including molecular markers e.g. genomic DNA markers, mRNA/miRNAs/lncRNAs/circRNAs markers, epigenetic markers, proteomics, and metabolics, could serve as potential biomarkers in outcome of burn patients [143]. The advantage of these non-inflammatory response markers could be that they can predict for instance sepsis in burn patients, as markers of the post-burn inflammatory response do not reflect the severity of the infection.

Analyses of changes in genetic processes in the skin during burns revealed three potential novel diagnostic markers in blood from burn patients. Three Hub genes MCEMP1, MMP9, and S100A12 were shown to be significantly elevated in burn patients and were suggested as key blood biomarkers that can be used to identify skin damage in burn patients [144].

Furthermore analyses of miRNAs that are potentially involved in early burn-response gene regulation have shown that miRNA-497-5p, which regulates skin cell regeneration, was downregulated in tissue and in dermal interstitial fluid (dISF) of burned skin. Therefore, further examination of miRNA-497-5p as a biomarker for the severity of burns is suggested [145].”

Round 2

Reviewer 1 Report

Thank you for the revised version of the manuscript. I think my comments and questions were addressed.

Reviewer 2 Report

 I am satisfied with the revised version of the paper. I recommend this paper to be published in this journal.